



# How robust are stratospheric age of air trends from different reanalyses?

Felix Ploeger[1,2], Bernard Legras[3], Edward Charlesworth[1], Xiaolu Yan[1], Mohamadou Diallo[1], Paul Konopka[1], Thomas Birner[4], Mengchu Tao[1], Andreas Engel[5], and Martin Riese[1]

[1]Institute for Energy and Climate Research: Stratosphere (IEK–7), Forschungszentrum Jülich, Jülich, Germany.
[2]Institute for Atmospheric and Environmental Research, University of Wuppertal, Wuppertal, Germany.
[3]Laboratoire de Météorologie Dynamique, UMR8539, IPSL, UPMC/ENS/CNRS/Ecole Polytechnique, Paris, France.
[4]Meteorological Institute, Ludwig-Maximilians Universität München, München, Germany.
[5]Institute for Atmospheric and Environmental Sciences, Goethe-University Frankfurt, Frankfurt, Germany.

*Correspondence to:* Felix Ploeger (f.ploeger@fz-juelich.de)

**Abstract.** An accelerating Brewer-Dobson circulation (BDC) is a robust signal of climate change in model predictions but has been questioned by trace gas observations. We analyze stratospheric mean age of air and the full age spectrum as measures for the BDC and its trend. Age of air is calculated with the Chemical Lagrangian Model of the Stratosphere (CLaMS) driven by ERA–Interim, JRA–55 and MERRA–2 reanalysis data to assess the robustness of the representation of the BDC in current gen-

eration meteorological reanalyses. We find that climatological mean age significantly depends on the reanalysis, with JRA–55 showing the youngest and MERRA–2 the oldest mean age. Consideration of the age spectrum indicates that the older age for MERRA–2 is related to a stronger spectrum tail, likely related to weaker tropical upwelling and stronger recirculation. Season-ality of stratospheric transport is robustly represented in reanalyses, with similar mean age variations and age spectrum peaks. Long-term changes over 1989–2015 turn out to be similar for the reanalyses with mainly decreasing mean age accompanied

by a shift of the age spectrum peak towards shorter transit times, resembling the forced response in climate model simulations to increasing greenhouse gas concentrations. For the shorter periods 1989–2001 and 2002–2015 age of air changes are less robust. Only ERA–Interim shows the hemispheric dipole pattern in age changes during 2002–2015 as viewed by recent satellite observations. Consequently, the representation of decadal variability of the BDC in current generation reanalyses appears less robust and a major uncertainty of modelling the BDC.

## 1   Introduction

The global circulation of the stratosphere, known as the Brewer-Dobson circulation (Brewer, 1949; Holton et al., 1995), is a crucial factor controlling lower stratospheric composition with radiatively active trace gases and therefore plays an important role for the Earth's radiation budget and for climate. Hence, a realistic representation of the BDC is a prerequisite for reliable climate model predictions. However, current climate models and observations disagree regarding long-term changes of the

BDC (e.g., Waugh, 2009). Climate models simulate a strengthening and accelerating circulation which is not evident from observations, representing a major uncertainty in current model predictions (see Butchart, 2014, for a recent review). The



BDC is characterised by the slow upwelling in the tropics from the troposphere across the Tropical Tropopause Layer (TTL) into the stratosphere followed by poleward motion in the stratosphere and downwelling at middle and high latitudes. From the perspective of tracer transport the BDC includes a residual mean mass circulation and additionally two-way eddy mixing which only causes net transport on tracers but not mass (e.g., Plumb, 2002). The residual mean mass circulation may be separated into two different main branches (Birner and Bönisch, 2011). A shallow circulation branch transports air masses to middle latitude regions at lower levels above the tropopause within a few months to about two years. A deep circulation branch, on the other hand, causes transport deep into the stratosphere and downwelling at high latitudes on a time scale of several years (Birner and Bönisch, 2011). The stratospheric BDC is mechanically driven by transfer of momentum from breaking atmospheric waves to the zonal flow (Haynes et al., 1991; Holton et al., 1995). The shallow circulation branch is mainly driven by synoptic and planetary scale waves and the deep branch is mainly driven by planetary waves propagating deep into the stratosphere (e.g., Plumb, 2002). Results from an idealised model suggest that the strength of the shallow circulation branch is largely controlled by the strength of wave sources in the troposphere (Gerber, 2012). The strength of the deep circulation branch, on the other hand, appears largely controlled by the propagation conditions for waves in the stratosphere.

As the BDC is a zonal mean circulation and related wind velocities are small residuals, directly measuring the circulation is not possible (e.g., Butchart, 2014). A common measure of the BDC is the age of air, the time scale for transport through the stratosphere. By definition, age of air measures the speed of the circulation. Strictly, due to atmospheric mixing processes on a multitude of scales a stratospheric air parcel is not characterised by a single transport time scale but by a transit time distribution. This transit time distribution is termed the age of air spectrum (Hall and Plumb, 1994). Commonly used reference surfaces for measuring the transit time are the tropical tropopause or the tropical surface (e.g., Waugh and Hall, 2002). The first moment of the age spectrum distribution, termed the mean age of air, can be estimated from observations of specific trace gas species with linearly increasing mixing ratios in the troposphere, as is approximately the case for $CO_2$ and $SF_6$ (Hall and Plumb, 1994). Similar to a tracer, mean age is affected by both the residual mean mass circulation and atmospheric mixing processes (e.g., Neu and Plumb, 1999; Garny et al., 2014), complicating an unambiguous interpretation of mean age changes in terms of processes. The full age spectrum allows a clearer interpretation, but its deduction from observations is much more challenging and usually hinges on simplifying assumptions like stationarity of the flow or a specific parameterization of the age spectrum shape (e.g., Andrews et al., 1999; Schoeberl et al., 2005). Ongoing modelling studies suggest promising new methods for deducing the age spectrum based on either an improved parametric approach (Hauck et al., 2018) or an inversion approach (Podglajen and Ploeger, 2018). Applicability to measurement data, however, remains to be shown.

The BDC is characterised by variability on very different time scales. Seasonal variability in the BDC is caused by seasonality in wave driving, being stronger in boreal than austral winter due to larger wave excitation by orography in the Northern Hemisphere (NH). As a result, tropical upwelling and related extratropical downwelling (in the NH) maximise in boreal winter (e.g., Yulaeva et al., 1994). On inter-annual time scales, the BDC is significantly modulated by the stratospheric Quasi-Biennial Oscillation (QBO) (e.g., Baldwin et al., 2001), by the El Niño Southern Oscillation (ENSO) (e.g., Calvo et al., 2010; Diallo et al., 2018), and by stratospheric aerosol injected by volcanic eruptions (e.g., Diallo et al., 2017). In the long term, climate models simulate a robust strengthening of the BDC, resulting in an increase of the tropical upwelling mass flux as well


as in a global decrease of stratospheric mean age of air over the last decades and into the future (e.g., Butchart et al., 2010; Butchart, 2014). In contrast, estimates of mean age of air from balloon observations of long-lived trace gas species in the NH middle latitudes above about 24 km show no significant long-term trend (Engel et al., 2009). Additional recent balloon measurements corroborate this result (Engel et al., 2017). A recent, improved analysis of the same balloon observations confirms
the insignificant (weakly positive) mean age trend above about 24 km, but shows a negative mean age trend below (Ray et al., 2014). Indications for a negative mean age trend in the lowermost stratosphere around the year 2000 have also been found from aircraft observations of mean age (Bönisch et al., 2011). Hence, in particular in the stratosphere above about 24 km climate models results are not consistent with observations regarding trends in the BDC.

The strengthening BDC trend in models is clearly related to greenhouse gas induced tropospheric warming. This warming
strengthens the subtropical jets, thus shifts the critical levels for wave breaking upwards and equatorwards and intensifies the mechanical forcing of the BDC (e.g., Garcia and Randel, 2008; Shepherd and McLandress, 2011; Garny et al., 2011). A recent study by Oberländer et al. (2016) indicates that the BDC trend is consistent with the expansion of the troposphere and an associated upward shift of the tropopause with climate change.

Concerning BDC changes over decadal periods horizontal circulation shifts in the latitudinal direction also cause important
effects. This has recently been shown for a southward circulation shift during 2002–2012 which likely caused a hemispheric dipole mean age change pattern (age increase in NH, decrease in SH) as observed by the Michelson Interferometer for Passive Atmospheric Sounding (MIPAS) satellite instrument (Stiller et al., 2017). Ensembles of climate model simulations may include mean age change patterns which are more complex than a global decrease with even some resemblance to MIPAS observations (Garfinkel et al., 2017). Hence, to simulate observed past BDC changes it is important to correctly represent variability on inter-
annual to decadal time scales in the models. For interpreting mean age trends, consideration of atmospheric mixing processes turns out to be particularly important (e.g., Ray et al., 2010; Ploeger et al., 2015a; Dietmüller et al., 2017).

Meteorological reanalyses combine a global weather forecast model with atmospheric observations through a data assimilation system to provide an optimal estimate of the atmospheric state from the past to present. These reanalyses are based on an unchanged forecast model version and assimilation system to minimise artificial changes in the state variables due to
changes in the reanalysis system. However, as the observational data sets included in the assimilation change over time abrupt state changes may still occur to some degree, rendering reanalysis based trend studies challenging. Considering more than one reanalysis in such studies increases the reliability of results considerably. The Stratosphere-troposphere Processes And their Role in Climate (SPARC) Reanalysis Inter-comparison Project (S–RIP) aims at an inter-comparison of the current generation reanalysis products, as described by Fujiwara et al. (2017). The present paper contributes to this project by comparing the rep-
resentation of the stratospheric BDC in the three most modern reanalysis products: (i) ERA–Interim from the European Centre for Medium-term Weather Forecasts (ECMWF), (ii) JRA–55 from the Japanese Meteorological Agency, and (iii) MERRA–2 from the National Aeronautics and Space Administration (NASA).

Past analyses have shown that modern reanalyses provide an improved representation of the BDC (e.g., Diallo et al., 2012; Monge-Sanz et al., 2012) compared to older reanalysis products like ECMWF's ERA–40 reanalysis (e.g., Monge-Sanz et al.,
2007). In particular, ERA–Interim has been shown to combine a negative mean age trend throughout most regions of the strato-



sphere with a weakly positive age trend in the NH above about 24 km, similar to existing balloon observations (Ploeger et al., 2015a). Moreover, the hemispheric dipole age trend pattern as observed by MIPAS during 2002–2012 turns out to be reproduced, at least qualitatively, by ERA–Interim driven simulations (Ploeger et al., 2015b). The robustness of these results concerning the representation of the BDC in different reanalyses, however, is an open question.

A very recent study by Chabrillat et al. (2018) compares the BDC in various reanalyses by using a kinematic transport model (Belgian Assimilation System for Chemical ObsErvations, BASCOE) within the scope of the S–RIP project. The results of the present paper here are based on a diabatic transport model (for further details see Sect. 2) and therefore complement the study by Chabrillat et al. (2018) regarding the representation of vertical transport. The main goal of our paper is to assess the robustness of the climatology and seasonality of the BDC as well as its trends in current generation reanalyses, as imprinted on

stratospheric age of air. For that reason we calculate and analyse mean age of air as well as the full time-dependent stratospheric age spectrum, which has not been used for a model inter-comparison of the BDC, hitherto.

The modelling method (transport model, age of air diagnostics) is described in Sect. 2. In Sect. 3 the climatology and seasonality of age of air from different reanalyses is compared before considering trends in Sect. 4. Section 5 provides a comparison to existing observational mean age estimates and Sect. 6 discusses the results, in particular against the results from

the complementary model study of Chabrillat et al. (2018).

## 2 Data and Method

The Chemical Lagrangian Model of the Stratosphere (CLaMS) is a Lagrangian model for calculating transport and chemistry for trace gas species based on the motion of 3D forward trajectories and an additional parameterised representation of atmospheric small-scale mixing processes (McKenna et al., 2002). Parameterised mixing in the model is driven by deformations

in the large-scale flow, such that in regions of large flow deformations strong mixing occurs (Konopka et al., 2004). Model transport is calculated in an isentropic vertical coordinate framework with potential temperature $\theta$ being the vertical coordinate throughout the stratosphere and upper troposphere, and with the cross-isentropic vertical velocity deduced from the total diabatic heating rate (of the respective reanalysis), including effects of radiative and turbulent heating, as well as latent heat release. Further details about the CLaMS model set-up used in this study can be found in Pommrich et al. (2014).

As described by Ploeger and Birner (2016), and briefly reviewed in the following, we calculate the age of air spectrum for each reanalysis from multiple tracer pulses and mean age from the spectrum. The age spectrum $G$ is a boundary value Greens function for the continuity equation of a conserved and passive trace gas species (e.g., Hall and Plumb, 1994; Holzer and Hall, 2000) and relates the trace gas mixing ratio $\chi(r,t)$ at location $r$ and time $t$ to the mixing ratio $\chi_0(t)$ at the boundary surface, where it is assumed to be uniform (e.g., Waugh and Hall, 2002)

$$\chi(r,t) = \int_0^\infty d\tau \, \chi_0(t-\tau) G(r,t,\tau). \tag{1}$$

Here, the integration is taken along transit time $\tau$, and the boundary surface is usually taken to be the tropical tropopause or the tropical surface, a particularly common choice in models. The age spectrum calculation applied in this study is set up





analogously to the approach described by Ploeger and Birner (2016), and similarly to the calculation in the GEOS climate model (Li et al., 2012a, b). The calculation method is based on $N = 60$ inert pulse tracers, approximating a delta distribution lower boundary condition $\chi_0^i(t) = \delta(t - t_i)$ with $i = 1, ..., 60$, defining tracer pulses at source times $t_i$. For such a set of pulse tracers, the age spectrum at transit time $\tau_i$ is related to the tracer mixing ratio of the $i$-th species via

$$G(r, t, \tau_i) = \chi_i(r, t),\tag{2}$$

and hence the age spectrum can be directly calculated from the pulse tracer mixing ratios in the simulation.

To approximate the delta distribution characteristics, pulse tracer mixing ratios are set to 1 in the lowest (orography following) model layer (approximately the boundary layer) in the tropics between 15°S–15°N for 30 days. The 60 different species are pulsed every other month (1st species in the first month, 2nd species in the third month, etc.), such that after 10 years of

simulation all species have been pulsed. Hence, after every 10 years the first species is reset to zero and pulsed again. This Boundary Impulse (time-)Evolving Response (BIER) method (Ploeger and Birner, 2016) resolves the age spectrum along 10 years of the transit time axis with a bin size of 2 months.

Mean age of air is then calculated as the first moment of the age spectrum

$$\Gamma(r, t) = \int_0^\infty d\tau\, \tau\, G(r, t, \tau).\tag{3}$$

As the mean age strongly depends on the age spectrum tail, which in general shows an exponential decay after about 4–5 years (e.g., Li et al., 2012a), the effect of the finite age spectrum tail on mean age may be corrected by fitting an exponentially decaying function (e.g., Diallo et al., 2012; Ploeger and Birner, 2016). Hence, a corrected age spectrum may be defined by extrapolating the spectrum tail for transit times $\tau > \tau^* = 10$ using

$$G_{\text{corr}}(r, t, \tau) = G(r, t, \tau^*)\, e^{-\frac{(\tau - \tau^*)}{\xi(r,t)}},\tag{4}$$

with $G(r, t, \tau^*)$ the age spectrum value at 10 years, and the tail decay time scale $\xi(r, t)$ (depending on location and time) being estimated from the exponential fit to the spectrum at transit times between 5–10 years. Being the probability distribution of transit times, the full age spectrum, in general, is normalised to unity. Due to the truncation of the simulated spectrum at 10 years, however, integration of the spectrum over transit time leads to a norm less than one (see Fig. 6)

$$N = \int_0^\infty d\tau\, G(r, t, \tau) < 1.\tag{5}$$

Including the exponential tail correction improves the normalisation, but small differences to unity remain due to the finite resolution along the transit time axis. On the one hand, including the correction generally improves comparisons of mean age to observations. On the other hand, for MERRA–2 age spectra show a much stronger tail without a clear exponential decay over 10 years in some cases (see Sect. 5), violating the necessary assumption for the finite tail correction. Hence, for most parts of the analysis we simply consider the finite tail age spectra over 10 years without including the tail correction. This simplifies



interpretation of the comparison of different reanalyses by only considering the resolved part of the age spectrum. Effects of the unresolved tail and the finite tail correction are further discussed in Sect. 5.

Two other age spectrum based transport diagnostics are the modal age, which is the transit time of the maximum age spectrum peak, and the age spectrum width

$$\Delta(r,t) = \sqrt{\frac{1}{2} \int\limits_{0}^{\infty} \mathrm{d}\tau \, [\tau - \Gamma(r,t)]^2 \, G(r,t,\tau)}. \tag{6}$$

The spectrum width is strongly influenced by long transit times in the spectrum tail and therefore is usually considered a measure for the strength of recirculation (e.g., Li et al., 2012a). The modal age, on the other hand, can be interpreted as a measure of the residual circulation in the tropics and in the winter stratosphere as it is closely related to the residual circulation transit time in these regions. As additional diagnostics for the interpretation of processes affecting zonal mean mean age we

consider residual circulation transit times (RCTT, Birner and Bönisch, 2011) and the net mixing effect on mean age ("ageing by mixing", Garny et al., 2014) in Sect. 6. RCTT is the transit time of a (hypothetical) air parcel if it was transported by the residual circulation alone and, by definition, solely includes effects of the residual circulation (Birner and Bönisch, 2011). For this paper, RCTTs are calculated with the CLaMS trajectory module and using the zonal mean diabatic residual circulation in isentropic coordinates $(\overline{v}^*, \overline{Q}^*)$ (Ploeger et al., 2015b). Here, Q denotes the diabatic heating rate, v the meridional velocity component,

overlined quantities denote mass-weighted zonal averages and primed quantities the respective fluctuations therefrom (e.g., Andrews et al., 1987, Ch. (9.4)), hence $\overline{v}^* = (\overline{\sigma v})/\overline{\sigma}$ and $\overline{Q}^* = (\overline{\sigma Q})/\overline{\sigma}$.

In addition, zonal mean mean age $\overline{\Gamma}$ is affected by mixing processes in the atmosphere. The local effect of this mixing, the local eddy mixing tendency, is represented in the zonal mean (isentropic) tracer continuity equation for mean age by the divergence $\mathcal{M}$ of a 2D mixing flux vector (e.g., Andrews et al., 1987, Eq. (9.4.21)). The net mixing effect on zonal mean

mean age, the ageing by mixing, is then calculated by integrating the local eddy mixing tendency $\mathcal{M}$ along residual circulation trajectories $x(t)$ (Ploeger et al., 2015b)

$$\overline{\Gamma}(x,t) = \tau_{\mathrm{RCTT}}(x,t) + \int\limits_{t_0}^{t} \mathcal{M}(x(t')) \, \mathrm{d}t'. \tag{7}$$

Here, $\tau_{\mathrm{RCTT}}$ is the RCTT, for transport from the 340 K isentropic surface in the tropics (30°S–30°N). As proposed by Garny et al. (2014) and recently further evidenced by Dietmüller et al. (2017), the net eddy mixing effect can be well approximated by the difference between mean age and RCTT. We apply this approximation in the following without explicitly calculating the mixing

effect (see Sect. 6). For further details about the RCTT and ageing by mixing calculation see Ploeger et al. (2015b).

The CLaMS simulations are driven with horizontal winds and diabatic heating rates from the three most recent reanalysis data sets ERA–Interim, JRA–55 and MERRA–2. The simulations for ERA–Interim and JRA–55 both start on 1 January 1979, whereas the MERRA–2 simulation starts on 1 January 1980. Due to the specific pulse tracer set-up described above it takes 10 years of simulation until all pulse tracers have been set and the age spectrum can be evaluated. To enable age of air analysis already from 1979 on (from 1980 on for MERRA–2), for comparison with balloon-borne mean age measurements in Sect. 5,

a 10 year long model spin-up is carried out by repeating conditions of the first simulation year. However, the simulated age of



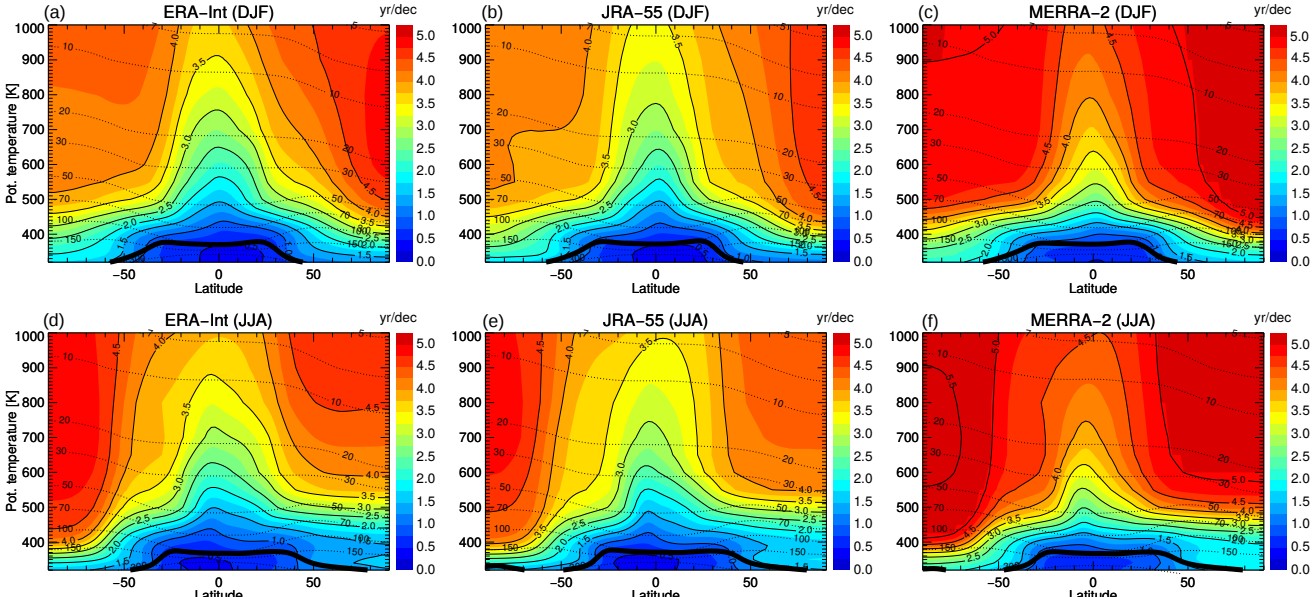

**Figure 1.** Mean age climatology (1989–2015) for December–February (DJF, top) and June–August (JJA, bottom) for ERA-Interim (left), JRA–55 (middle), and MERRA-2 (right). Thin solid black lines highlight particular mean age contours, thin dashed black lines show pressure levels in hPa, and the thick black line is the (lapse rate) tropopause (calculated from each reanalysis following WMO, 1957).

air before 1989 includes the effect of the spin-up and most parts of the analysis presented here are restricted to the time period 1989–2015. For MERRA–2, the year 1989 still includes a very weak spin-up effect, but without any effect on our conclusions, as all results of the paper could be analogously derived for the 1990–2015 period (not shown).

The three different reanalyses used here have been recently described by Fujiwara et al. (2017), with further details given by Dee et al. (2011) for ERA–Interim, by Kobayashi et al. (2015) for JRA–55, and by Gelaro et al. (2017) for MERRA–2. For driving the CLaMS model simulations, reanalysis horizontal winds and diabatic heating rates from the reanalysis forecast are used on native model levels and with a horizontal resolution of $1 \times 1$ degrees in latitude and longitude. The age of air results from the different simulations have been finally interpolated to potential temperature levels (same for all reanalyses) and monthly zonal mean climatologies have been created.

## 3  Seasonal variations in age of air

Climatological mean age data (1989–2015) from the three reanalyses for boreal winter (December–February, DJF) and summer (June–August, JJA) are compared in Fig. 1. Throughout most parts of the stratosphere JRA–55 shows the youngest mean age and MERRA–2 shows the oldest mean age. Values for ERA–Interim lie in between. Only in the tropical lower stratosphere



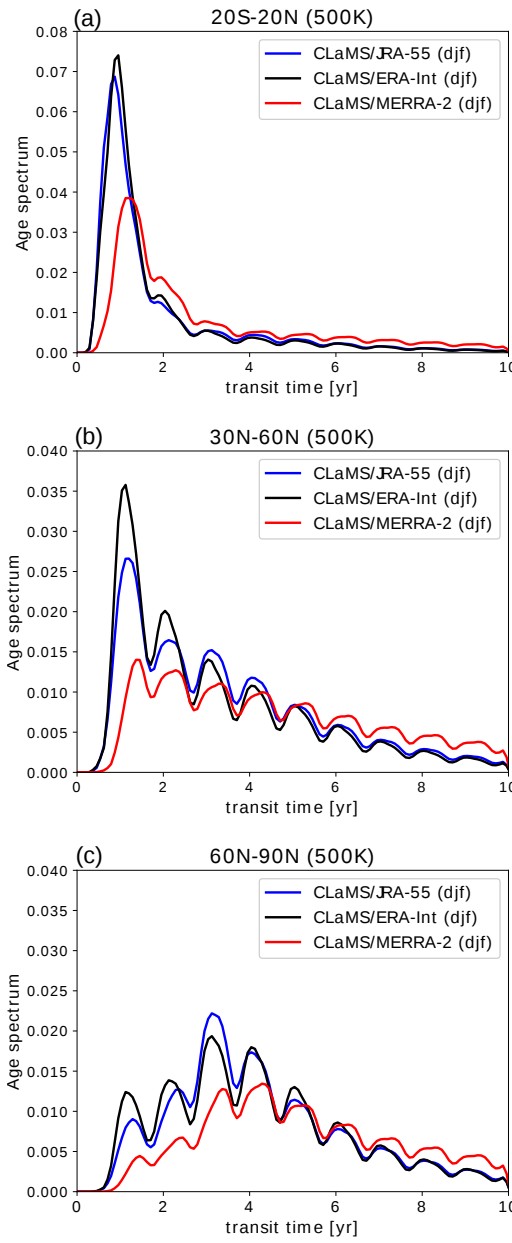

**Figure 2.** Age spectrum on the 500 K potential temperature surface for December–February in (a) the tropics between 20°S–20°N, (b) middle latitudes between 30°N–60°N, and (c) polar regions between 60°N–90°N. Different colours represent different reanalyses (ERA–Interim black, JRA–55 blue, MERRA–2 red). (Note the different y-axis range for tropical and extratropical spectra.)





below about 700 K potential temperature (about 20 hPa) ERA–Interim shows the youngest mean age of the three reanalyses, which is consistent with too strong tropical upwelling as documented in the literature (e.g., Dee et al., 2011).

MERRA–2 mean age is substantially older compared to the other reanalyses, with largest relative differences in the tropical stratosphere (e.g., about 2.5 yr for MERRA–2 at about 500 K (about 50 hPa) compared to about 1.5 yr for ERA–Interim). Also

at high latitudes MERRA–2 mean age is clearly older, reaching maximum values of more than 5.5 yr in the Southern polar vortex compared to less than 5 yr for ERA–Interim and JRA–55.

Despite these differences in climatological average mean age values, seasonal variations (as estimated from December–February to June–August differences) are very similar in the three reanalyses. The youngest air is found in the tropical stratosphere during boreal winter, when BDC upwelling maximises. On the other hand, the oldest air occurs in the high-

latitude stratosphere during wintertime and is related to the strongest downwelling within the deep BDC branch in that region and season. Furthermore, a flushing of the summertime lower stratosphere with young air masses from the tropics (e.g., Hegglin and Shepherd, 2007; Bönisch et al., 2009) is evident in all reanalyses. This summertime flushing shows a robust hemispheric asymmetry, being stronger in the NH compared to the SH (Konopka et al., 2015).

More details about stratospheric transport can be observed in the age spectrum, presented in Fig. 2 for the tropics, middle,

and high latitudes on the 500 K potential temperature surface. The tropical age spectra show an almost unimodal shape with a clear peak at short transit times, and only very weak additional peaks and a decaying tail at larger transit times. In middle and high latitudes the age spectra show distinct multiple peaks caused mainly by the seasonality of transport into the stratosphere (e.g., Reithmeier et al., 2007; Li et al., 2012a; Diallo et al., 2012; Ploeger and Birner, 2016). These multiple spectrum peaks are robustly found for all reanalyses, again indicating robustness in the representation of seasonal variations in stratospheric

transport in modern reanalyses. In general, ERA–Interim and JRA–55 age spectra are very similar for all regions, but MERRA–2 spectra differ more substantially. In particular, for MERRA–2 the secondary peaks at older ages are delayed by a few months compared to ERA–Interim and JRA–55, indicating again slower transport for MERRA-2. At high latitudes, the modal age (transit time of the maximum spectrum peak) for MERRA–2 may occur with a delay of more than a year (e.g., Fig. 2c). In all regions the modal peak is shifted to larger transit times (older ages) for MERRA–2. In particular, the spectrum tail is much more

pronounced for MERRA–2 (e.g., Fig. 2f), with age spectrum values more than twice as large compared to ERA–Interim and JRA–55 at transit times larger than about 8 years. Hence, there is a substantially larger fraction of very old air for MERRA–2 than for ERA–Interim and JRA–55.

The complete global view of the age spectrum in the lowest stratosphere at 400 K is presented in Fig. 3. The 400 K isentrope has been chosen as a representative level for the shallow BDC branch. The robust representation of transport seasonality is

again evident from the similar occurrence of the multiple spectrum peaks. In particular, the modal peak (modal age) of tropical age spectra is shifted to younger ages in boreal winter, consistent with faster wintertime BDC upwelling. The strong flushing of the Northern hemisphere (Bönisch et al., 2009), and to a lesser degree also of the Southern hemisphere, is clearly visible in the extension of the modal peak (white diamonds) from the tropics into the summer hemisphere in all reanalyses. This extension of the tropical young air signal deep into the summer hemisphere is consistent with the general understanding of a less isolated

tropics and stronger isentropic mixing causing horizontal exchange between tropics and extratropics during summer.





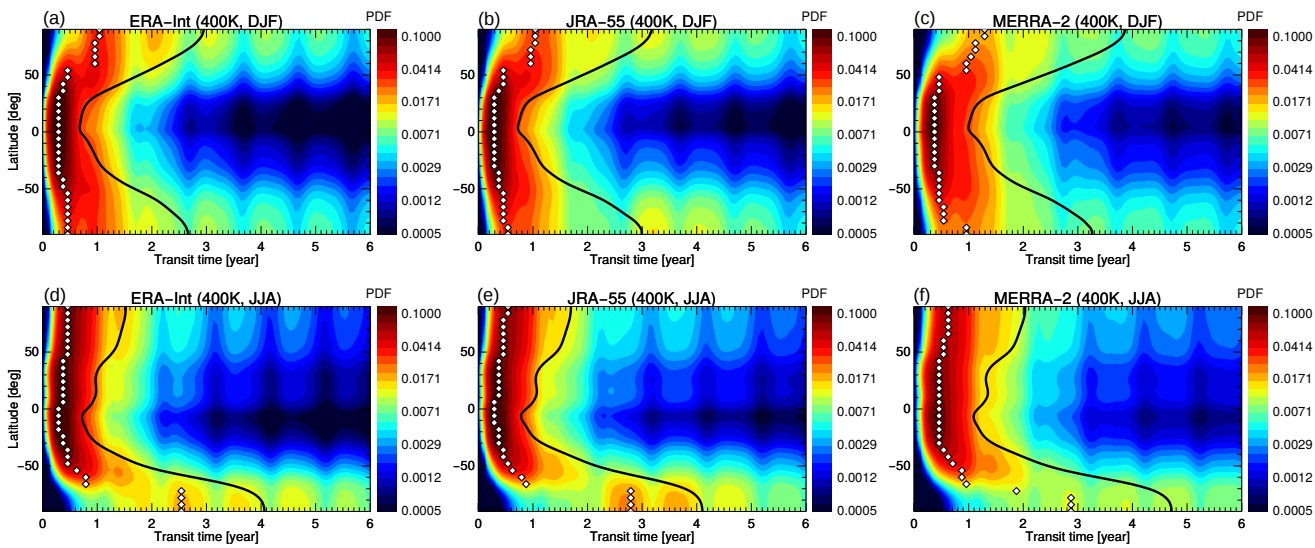

**Figure 3.** Age spectrum at 400 K for December–February (top) and June–August (bottom) for ERA–Interim (left), JRA–55 (middle), and MERRA–2 (right). Shown are climatological values for 1989–2015. The black line shows mean age, the white diamonds show modal age.

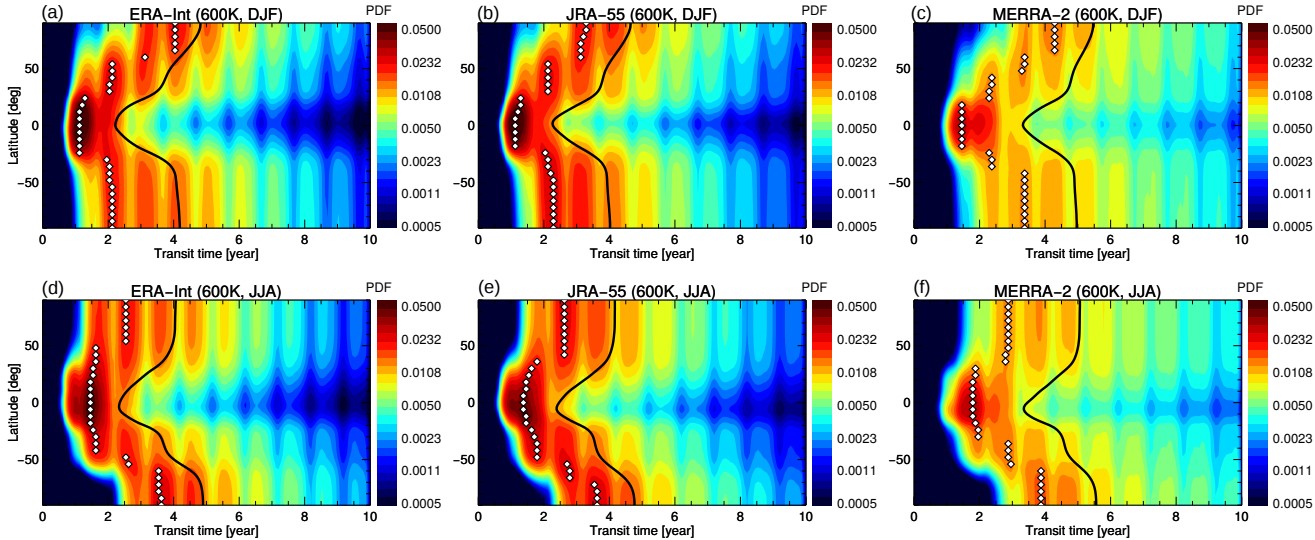

**Figure 4.** Same as Fig. 3, but at the 600 K potential temperature surface.

The subtropics in Fig. 3 are characterised as transition regions between approximately monomodal tropical age spectra and middle and high latitude spectra with distinct multiple peaks. The contrast between tropical and extratropical age spectra is strongest in the respective winter hemisphere, indicative of a stronger subtropical jet and related transport barrier compared





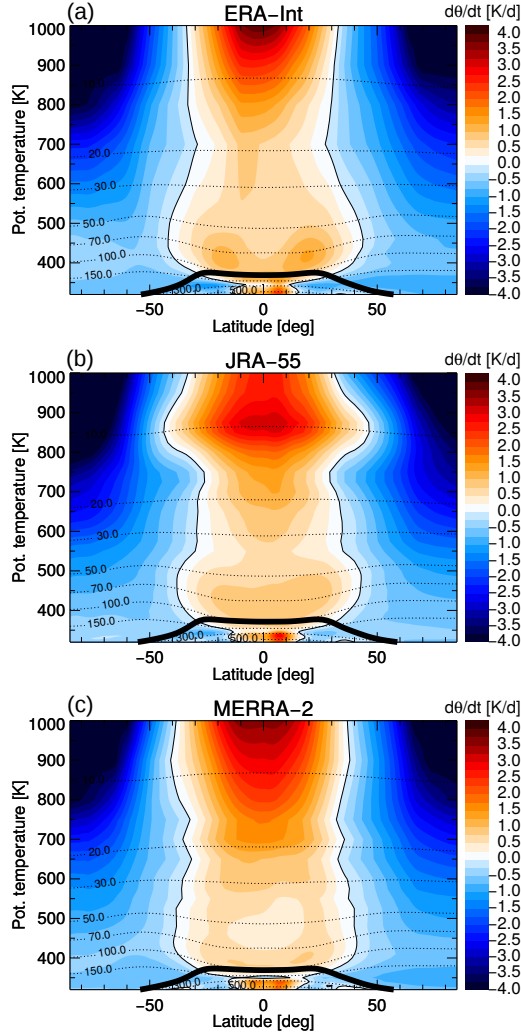

**Figure 5.** Cross-isentropic vertical velocity dθ/dt from the total diabatic heating rate for ERA–Interim (a), JRA–55 (b), and MERRA–2 (c). Shown are climatological annual mean distributions for 1989–2015. Thin black lines show pressure contours, the thick black line the (lapse rate) tropopause.

to the summer hemisphere. For MERRA–2, the tropics-extratropics transition is more dilute, indicating stronger exchange between tropics and middle latitudes in the lowest stratosphere. This stronger exchange likely results in a stronger recirculation of extratropical aged air masses into the tropics, causing older air throughout the stratosphere.

Very similar results emerge from the comparison of the age spectra at higher levels in the stratosphere (here 600 K in Fig. 4), where the deep BDC branch dominates transport. The seasonality in tropical upwelling (faster in boreal winter) as well



as in subtropical transport barriers (stronger in boreal winter) is robustly represented in the different reanalysis data sets. In particular the older modal age for MERRA–2 in the tropics, already evident at 400 K (Fig. 3), becomes even clearer at 600 K.

This shift of the tropical modal age to longer transit times is related to a slower tropical upwelling for MERRA–2, consistently emerging also from comparison of reanalysis climatological total diabatic heating rates shown in Fig. 5. In particular
throughout the Tropical Tropopause Layer (e.g., Fueglistaler et al., 2009a) and up to about 500 K the MERRA–2 heating rates are lower than for the other reanalyses (Fig. 5). Below the tropopause around the level of zero radiative heating (around 360 K) there is even a cooling layer evident in the MERRA–2 heating rates causing a "bottleneck" structure in annual mean upwelling. The differences in heating rates could be related to ozone differences between the reanalyses (e.g., Davis et al., 2017). ERA–Interim uses prescribed climatological ozone fields for the radiative calculations, including a potential high bias
in ozone concentrations in the tropical lower stratosphere (e.g., Fueglistaler et al., 2009b). JRA–55, on the other hand, uses prescribed time-varying ozone fields from a model simulations (Kobayashi et al., 2015), and MERRA–2 uses interactive ozone for radiation and heating rate calculations (e.g., Wargan et al., 2018).

Regarding the age spectrum, MERRA–2 further shows a weaker gradient between tropical and extratropical age spectra compared to the other reanalyses also at 600 K (Fig. 4). This weaker tropics–extratropics contrast indicates stronger exchange
between tropics and extratropics and a weaker subtropical transport barrier. Furthermore, the MERRA–2 age spectra at 600 K show a more pronounced spectrum tail, as already noted. Differences in the age spectrum tail at transit times larger than about five years cannot be caused by the differences in the diabatic circulation (in Fig. 5) alone, because related transit times along the residual circulation are in general below about five years (e.g., Birner and Bönisch, 2011, Fig. 2). Therefore, the more pronounced spectrum tail for MERRA–2 compared to ERA–Interim and JRA–55 is likely a result of the stronger recirculation
of air into the tropics at lower levels. Recirculation into the tropics, particularly at low levels, causes air masses to circulate several times through the stratosphere within the BDC before sinking back into the troposphere and significantly increases the age (e.g., Neu and Plumb, 1999; Garny et al., 2014). Therefore, an increased recirculation enhances the fraction of aged air masses and hence the tail of the age spectrum (e.g., Li et al., 2012a).

Figure 6 further compares the characteristics of the spectrum tail between the three reanalyses by showing the spectrum
normalisation (colour shading) and the tail decrease time scale (blue contours). The spectrum norm was calculated as the integral of the age spectrum over 10 years (Eq. 5), and the tail decrease time scale as the exponential decrease rate of the spectrum tail at transit times larger than 5 years (Eq. 4). As the model age spectra are truncated after 10 years (see Sect. 2) a stronger spectrum tail will result in a less stringent normalisation condition and in a larger difference of the age spectrum norm from unity. Indeed, Fig. 6 shows that the MERRA–2 age spectra are less well normalised, with a norm below 0.8 throughout
large regions of the stratosphere, whereas for ERA–Interim and JRA–55 the norm is always above 0.9 (in the lower part of the stratosphere even above about 0.95). This lack in normalisation of MERRA–2 age spectra is clearly related to a much larger tail decrease time scale (global average 5.13 years) compared to ERA–Interim (2.84 years) and JRA–55 (2.91 years). Compared to the climate model simulated age spectra of Li et al. (2012a), with a global average tail decay time scale of 2.77 years, ERA–Interim and JRA–55 show comparable values while MERRA–2 differs substantially.



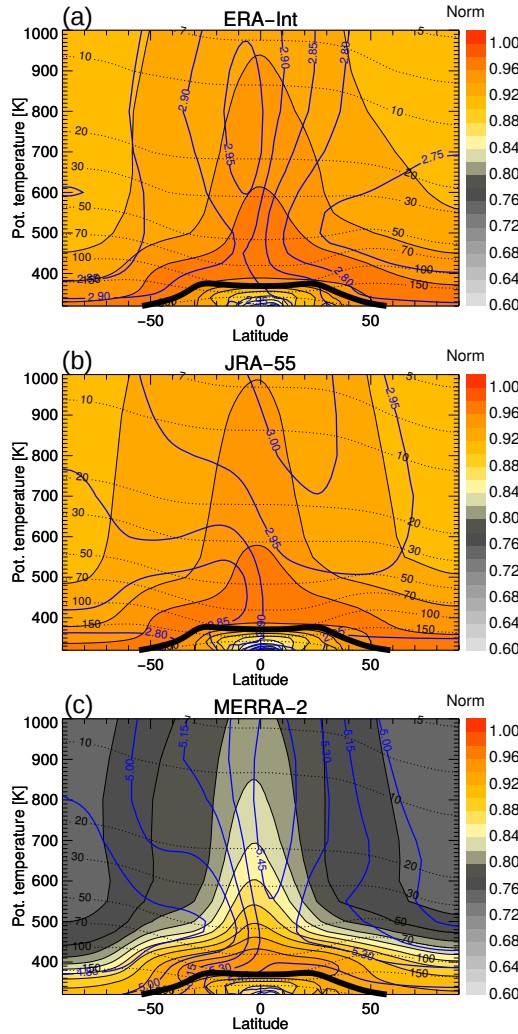

**Figure 6.** Norm of the climatological (1989–2015) age spectra for ERA–Interim (a), JRA–55 (b), and MERRA–2 (c). Blue contours show the climatological tail decay timescale (in years). Thin black dashed lines show pressure levels in hPa and the thick black line is the (lapse rate) tropopause.

## 4 Age of air trends

In the following, we analyse trends in age of air for the same periods as considered by Chabrillat et al. (2018) to simplify further comparison to their results in Sect. 6. We refer to 1989–2015 as "long-term" trends and to 1989–2001 ("pre–2000" in the following) and 2002–2015 ("post–2000") as "decadal" trends. It should be noted that even the long-term period spans only 27 years, which is relatively short compared to climate model simulation periods but is the longest period we can obtain from the reanalyses without including spin-up effects in the results (see Sect. 2).





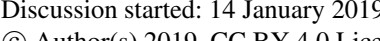

**Figure 7.** Long-term mean age trends for 1989-2015 (top), decadal trends for 1989-2001 (middle), and decadal trends for 2002-2015 (bottom), for ERA–Interim (left), JRA–55 (middle), and MERRA–2 (right). The trend significance, estimated in multiples of standard deviation σ, is shown as grey contours (2-σ contour thick grey, then decreasing in 0.2 steps as thin lines). Thin black solid contours show the climatological mean age distribution. Thin black dashed lines show pressure levels in hPa and the thick black line shows the (lapse rate) tropopause.

## 4.1 Long–term trends during 1989–2015

Long-term mean age trends for 1989–2015 are presented in Fig. 7. Overall, mean age trends are negative throughout most parts of the lower stratosphere below about 600 K (30 hPa) and for all three reanalyses. Hence, the representation of long-term changes in mean age in the lower stratosphere in current reanalysis is largely robust and indicates an accelerating shallow BDC branch. Mean age trends above about 600 K are not robust, with ERA–Interim showing positive trends in the NH whereas JRA–55 and MERRA–2 show negative trends. Hence, trends in the deep BDC branch appear not robustly represented in



current reanalysis data sets. The mean age decrease in the shallow BDC branch is in agreement with an acceleration of the residual circulation in reanalyses (Abalos et al., 2015; Miyazaki et al., 2016).

Closer examination of the 1989–2015 trends in Fig. 7 reveals detailed differences between the reanalyses. First, ERA–Interim shows an inhomogeneous pattern with strongest negative trends in the Southern subtropics, and even weakly positive trends in the NH above about 30 hPa. Second, JRA–55 provides a very homogeneous picture, showing negative mean age trends everywhere. Third, MERRA–2 shows again largely negative trends, maximising in the Southern subtropics similarly to ERA–Interim, and positive trends in the NH lowermost stratosphere below about 400 K.

Changes in the full age spectrum during the respective period can shed more light on the processes involved. Figure 8a–c shows age spectrum changes (colour shading) during the 1989–2015 period in the lowest stratosphere at 400 K and at all latitudes for the different reanalyses, overlaid with contours of climatological mean age (thick black lines). Remarkably, trends in the age spectra at young transit times are very consistent, particularly in the tropics, with all reanalyses showing an increase of the fraction of air masses with transit times younger than the modal age, and a simultaneous decrease in the air mass fraction with transit times just older than modal age. These changes indicate a shift of the spectrum peak (modal age) to younger transit times over time. Modal age can be related to the residual circulation transit time (e.g., Li et al., 2012a), particularly in the tropics and winter hemisphere stratosphere (Ploeger and Birner, 2016), and the shift of modal age to younger age in the tropics indicates an acceleration of the residual mean mass circulation.

This decrease of modal age emerges even more clearly at higher levels (e.g., 600 K in Fig. 9a–c). Independent of reanalysis and latitude, the spectrum peak shifts to younger transit times over the 1989–2015 period. Differences in mean age trends between the different reanalyses appear related to differences in the age spectrum tail. Clearly, the weakly increasing mean age in the NH above about 30 hPa (Fig. 7) in ERA–Interim is related to an increasing age spectrum tail at transit times older than about 4 years (Fig. 9a), which is absent in the other two reanalyses. Note that the 600 K potential temperature level is at the lower boundary of the region of increasing mean age in ERA–Interim, and that the described increase of the age spectrum tail becomes even clearer at levels above. However, 600 K has been chosen for consistency reasons with the later 2002–2015 trends, which show the mean age dipole change pattern only below about 20 hPa.

In agreement with the general shift of the age spectrum peak towards younger transit times during 1989–2015 the fraction of young air mass, younger than 6 months, robustly increases in the lower stratosphere in all three reanalyses (Fig. 10). Differences occur in the NH lowermost stratosphere, with the young air mass increase being strongest for JRA–55 and being absent for MERRA–2, consistent with the respective changes in mean age (Fig. 7). Note that the young air mass fraction is a particular robust diagnostic as it is independent of the truncation of the age spectrum to 10 years. Observable changes in the young air mass fraction are confined below about the 500 K potential temperature level. Above, the air is generally older than 6 months and the young air mass fraction with transit times shorter than 6 months almost vanishes. Changes in the old air mass fraction, older than 2 years, show negative changes in the tropical stratosphere above about 450 K (not shown), consistent with the shift of the age spectrum towards shorter transit time also at higher levels.



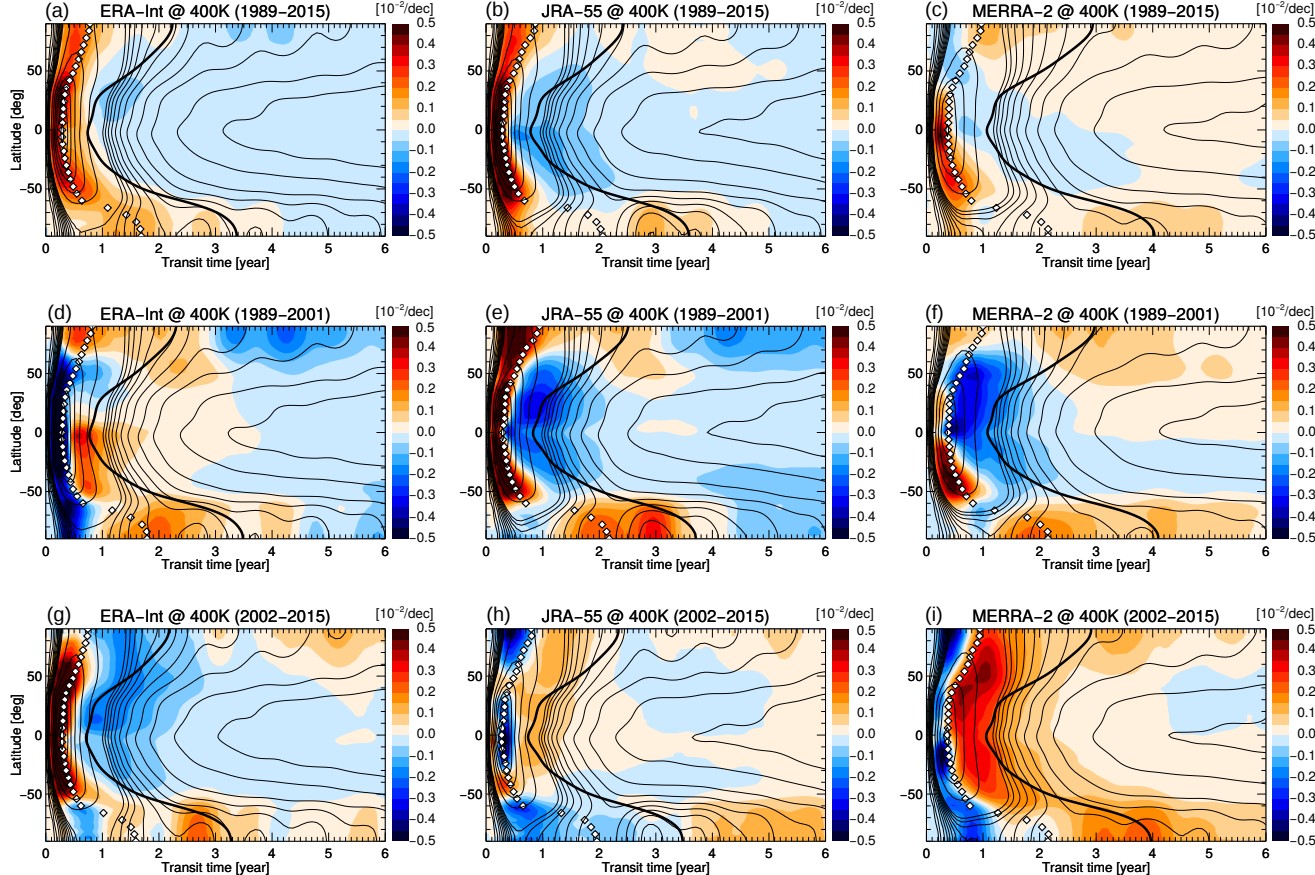

**Figure 8.** Age spectrum trend for at 400 K for ERA–Interim (left), JRA–55 (middle), and MERRA–2 (right), and for the periods 1989-2015 (top), 1989-2001 (middle), and 2002-2015 (bottom). The thin black contours show the climatological age spectrum, the thick black line climatological mean age, and the white diamonds climatological modal age.

## 4.2 Decadal changes during 1989–2001 and 2002–2015

The decadal mean age changes over shorter periods are much more diverse than the long-term trends. Furthermore, certain characteristics in these decadal changes depend critically on the start and end points of the period considered and should not be taken representative for long-term trends. Nevertheless, as observational data sets exist only for restricted periods and several past studies focused on circulation changes before and after the year 2000 we include a discussion of such decadal changes, in the following. The decadal periods considered (1989–2001 and 2002–2015) were chosen for better comparability with Chabrillat et al. (2018).

For 1989–2001, ERA–Interim and MERRA–2 show mean age trend patterns similar to the long 1989–2015 period (Fig. 7d and f). JRA–55, on the other hand, shows much stronger negative trends, particularly in the tropics (Fig. 7e). These strong


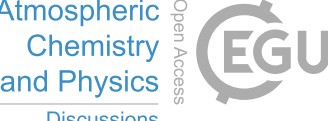


**Figure 9.** Same as Fig. 8, but at the 600 K potential temperature surface.

negative trends in JRA–55 are related to a stronger effect of the volcanic eruption of Mt. Pinatubo in June 1991, which increased mean age in the reanalysis in the lower stratosphere during 2-3 years following the eruption (Diallo et al., 2017). As the Pinatubo eruption and related ageing of air occurs close to the beginning of the trend period, 1989–2001 negative decadal mean age trends appear enhanced for JRA–55 (see Fig. 11b). More generally, the strong negative mean age trends in

5 the Southern hemisphere (SH) during the 1989–2001 period are likely related to the chemical and radiative effects of ozone depletion (Polvani et al., 2018). Reanalyses may include ozone depletion effects by assimilating observed temperatures even without having realistic ozone fields.

Mean age trends for the later period 2002–2015 are least consistent among the three reanalyses (Fig. 7g–i). MERRA–2 shows negative trends similar to the 1989–2015 and 1989–2001 periods. On the other hand, JRA–55 shows positive trends throughout

10 almost the entire stratosphere, with the exception of a small region in the NH lower stratosphere of insignificant trends. In contrast, ERA–Interim shows decreasing age in the lowest stratosphere, and a clear dipole pattern above with increasing age in the NH lower stratosphere, and decreasing age in the SH. This dipole pattern has been shown to be consistent with mean age




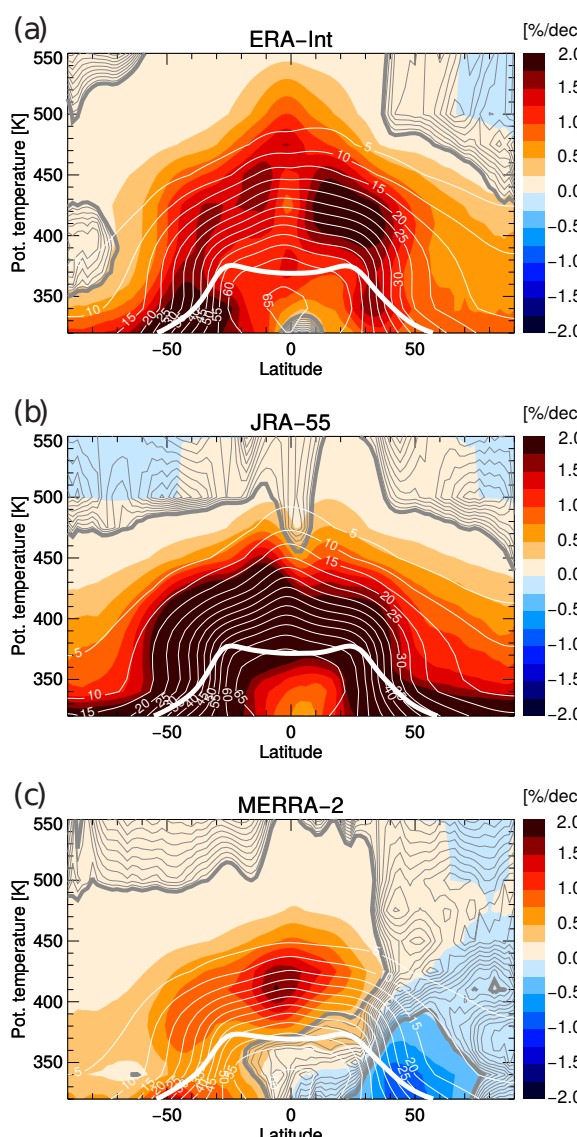

**Figure 10.** Trend in the young air mass fraction (transit time less than 6 months) during 1989–2015 for (a) ERA–Interim, (b) JRA–55, and (c) MERRA–2, in percent per decade. The white contours show the young air mass fraction climatology in percent, the thick white line shows the tropopause.





trends based on satellite observations (e.g., Haenel et al., 2015; Ploeger et al., 2015b) and observed HCl trends (Mahieu et al., 2014), although for the slightly different period 2002–2012. Recently, Stiller et al. (2017) related the dipole trend pattern to a southward shift of the global stratospheric circulation. Obviously, only ERA–Interim shows this dipole pattern in mean age trends for 2002–2015, whereas the other two reanalyses provide only very weak indications for slightly stronger ageing in the

NH (JRA–55) or a weaker age decrease in that region (MERRA–2).

As in the case of mean age, the decadal changes in the age spectrum over the shorter periods 1989–2001 and 2002–2015 are more diverse for the different reanalyses than the long-term 1989–2015 changes. At the lower potential temperature level of 400 K the spectrum peak may shift towards younger transit times (e.g., JRA–55 for 1989–2001) or towards older transit times (e.g., MERRA–2 for 2002–2015) and the spectrum tail may decrease or increase depending on the reanalysis (Fig. 8d–i).

At the upper level of 600 K, more consistent spectrum changes emerge. During the early period 1989–2001 all reanalyses show a shift of the spectrum peak towards younger transit times (Fig. 9d–f), consistent with decreasing mean age over this period. During the later period 2002–2015 MERRA–2 shows mainly a shift of the spectrum peak to shorter transit times (Fig. 9i), consistent with decreasing mean age (Fig. 7i). JRA–55, on the other hand, shows a shift of the spectrum peak towards longer transit times and an increase in the spectrum tail (Fig. 9h), resulting in positive mean age trends (Fig. 7h). ERA–Interim

shows a hemispheric asymmetric pattern (Fig. 9g). In the NH, the spectrum peak shifts towards longer transit times and the spectrum tail increases, resulting in increasing mean age during this period (Fig. 7g). In the SH opposite changes occur, with a spectrum peak shift towards shorter transit times and a decrease of the spectrum tail. Hence, the hemispheric asymmetric age spectrum changes in ERA–Interim for transit times less than 10 years are consistent with the hemispheric dipole trend in mean age. Clearly, spectrum changes in the tail at transit times of around 5 years appear critical for the positive mean age trend

in the NH. As 5 years is beyond the time scale of the residual circulation, mixing effects are likely involved in causing the hemispheric dipole pattern of mean age changes, as recently concluded from analysis of different diagnostics (Ploeger et al., 2015b; Stiller et al., 2017). Interestingly, age spectrum changes in the NH during 2002–2015 appear qualitatively consistent between ERA–Interim and JRA–55, showing decreasing values at transit times less than about 2 years and increasing values at transit times around 5 years. Hence, the more pronounced NH ageing in ERA–Interim compared to JRA–55 is the result of a

very subtle balance between changes in the age spectrum at short transit times (less than about 2 years) and in the tail (around 5 years).

## 5   Comparison to observations

To assess the reliability of the representation of the stratospheric BDC in different reanalyses, mean age of air is compared to mean age estimated from observations of the long-lived trace gas species $SF_6$ and $CO_2$. Figure 11a compares latitude sections

of mean age from reanalysis, air-borne in-situ observations (same data as shown in Waugh and Hall (2002) based on various measurements (Boering et al., 1996; Andrews et al., 2001; Elkins et al., 1996; Ray et al., 1999; Harnisch et al., 1996)), and MIPAS satellite observations of $SF_6$ (Stiller et al., 2012; Haenel et al., 2015) at 500 K potential temperature (20 km for the observations).



Overall, the reanalysis mean age lies within the uncertainty range of the observations. In the tropics, in-situ observed mean age is significantly lower than reanalysis age, with only ERA–Interim reaching similar values. MIPAS, however, shows much older mean age (above 2 years), similar to MERRA–2 values. Compared to the in-situ observations, age gradients in the subtropics are too weak for all reanalyses due to the higher tropical age values. In middle and high latitudes, ERA–Interim and

JRA–55 agree closely with $CO_2$ based mean age observations, whereas MERRA–2 agrees better with $SF_6$ based mean age.

As the reanalysis age spectrum is truncated at 10 years, the respective mean age is low biased by definition. This low-bias can be corrected by applying a correction for the finite age spectrum tail (see Sect. 2), and the corrected mean ages are also compared to the observations in Fig. 11a (dashed lines). For all reanalyses the tail correction increases mean age. This effect is moderate for ERA–Interim and JRA–55, such that the corrected reanalysis mean age remains within the observational

uncertainty range. For MERRA–2, in contrast, the effect of the tail correction is large, increasing mean age by more than 2 years at high latitudes. As a consequence, the tail-corrected MERRA–2 mean age is clearly out of the observational uncertainty range (the tail correction effects will be further discussed below).

A long-term observational mean age time series exists only for NH middle latitudes from balloon-borne measurements of $SF_6$ and $CO_2$ (Engel et al., 2009, 2017). Long-term time series of reanalysis mean age in NH middle latitudes (averaged between

40–50°N and 600–1200 K, approximately equivalent to the vertical layer for the observations of 30–5 hPa) are compared against this balloon observation time series in Fig. 11b. Clearly, the uncertainty arising from using different reanalyses is of similar magnitude as the uncertainty in the observations, such that no conclusion is possible which reanalysis scenario is most realistic. The mean age trends, however, substantially differ between the reanalyses with negative trends for JRA–55 ($-0.17 \pm 0.01$ yr/dec) and MERRA–2 ($-0.18 \pm 0.01$ yr/dec) and a positive trend for ERA–Interim ($0.07 \pm 0.01$ yr/dec), Hence,

ERA–Interim appears to agree best with the observed (non-significant) trend of $0.15 \pm 0.18$ yr/dec by Engel et al. (2017). A "best fit" to the observational time series results from applying the finite tail correction to ERA–Interim mean age, which is in remarkably good agreement to the observations, regarding absolute values, variability and trend (red dashed line).

Figure 11a shows a much stronger sensitivity to the tail correction for MERRA–2 mean age compared to the other reanalyses. This stronger sensitivity for MERRA–2 results from differences in the spectrum tail compared to the other two reanalyses (see

Fig. 2), with a substantially slower tail decrease for MERRA–2. Figure 12 illustrates a case of extreme differences between the reanalyses (June 1992 at 700 K potential temperature and 46°N). For ERA–Interim, age spectrum values decrease substantially over the 10 years of transit time (more than an order of magnitude), indicating a strongly decaying spectrum tail. For MERRA–2, in contrast, the tail decrease is much slower, leading to a substantially larger tail decay time scale and to a less strict normalisation, as already discussed above (Fig. 6). Caused by this larger tail decay time scale the finite tail correction applied

to the age spectra resolved over 10 years has a much larger effect for MERRA–2 compared to ERA–Interim and JRA–55. Figure 12 further shows that for MERRA–2 even the assumption of an exponentially decaying age spectrum tail after about 5 years, which has been found generally valid for observations (Ehhalt et al., 2004) and models (Li et al., 2012a), is violated in some cases. Hence, the exponential finite tail correction is not applicable for MERRA–2 age spectra over 10 years of transit time. For that reason, the inter-comparison presented in this paper is based on the uncorrected age spectrum.





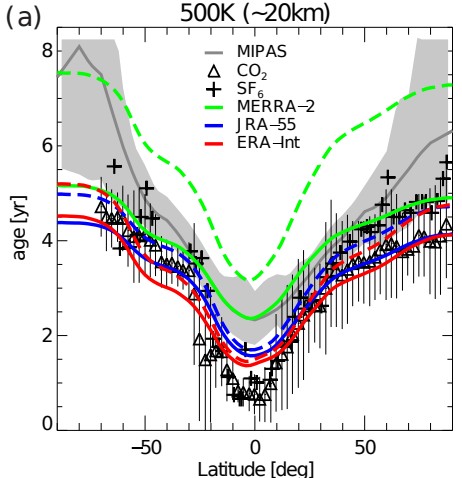

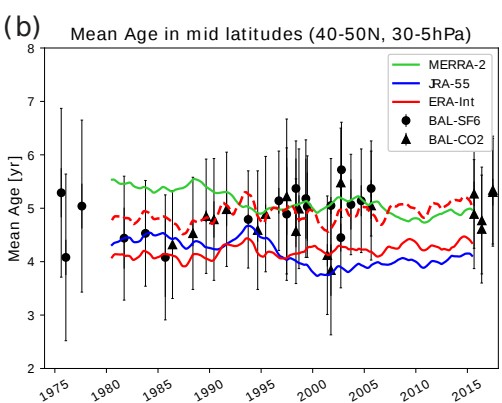

**Figure 11.** (a) Latitude section of mean age at 20 km from in-situ observations (e.g., Waugh, 2009) shown by black symbols, and from different reanalysis data sets at 500 K potential temperature (red: ERA-Interim; blue: JRA-55; green: MERRA-2), and from MIPAS satellite observations (grey). Dashed lines show reanalysis mean age including the correction for the finite age spectrum tail (see text). Grey shading shows the range between maximum and minimum MIPAS observations at each latitude. (b) Mean age time series in Northern middle latitudes (40°N–50°N and 30–5 hPa, 600–1200 K for reanalyses). Coloured lines show mean age from the different reanalyses (red dashed line corrected mean age for ERA–Interim). Black symbols show mean age from the balloon observations of Engel et al. (2017), with error bars representing the uncertainty of the observations.

# 6   Discussion

Potential long-term changes in the stratospheric Brewer-Dobson circulation and their relation to anthropogenically forced climate change have been a subject of intense debate over the last years. Climate models show a significant strengthening and acceleration of the BDC (Butchart, 2014), resulting in a global negative mean age trend (e.g., Butchart et al., 2010). Trace gas observations, on the other hand, show no indication for a significant long-term mean age trend (Engel et al., 2009, 2017).




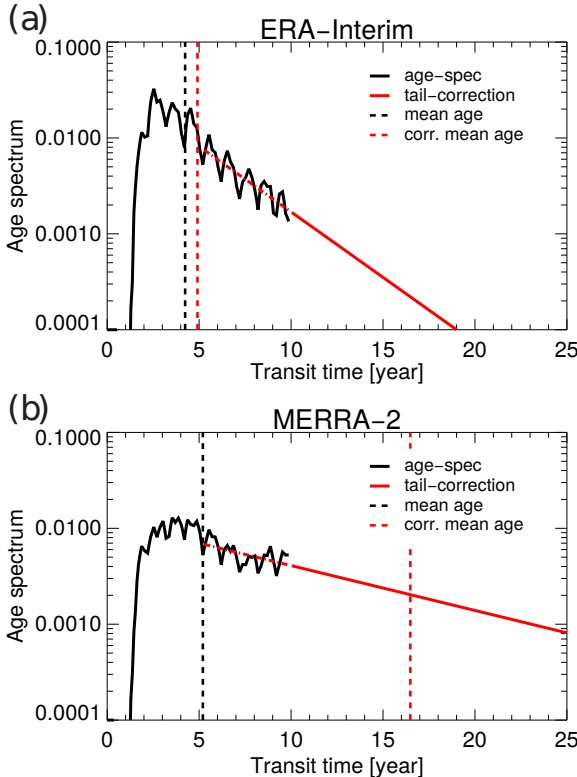

**Figure 12.** (a) Age spectrum at 46°N and 700 K potential temperature for June 1992 from ERA–Interim (a) and MERRA–2 (b). Red lines illustrate the correction method for the finite age spectrum tail by fitting an exponential function to the tail at transit times older than 5 years (see Sect. 2). The dashed red vertical line shows mean age for the tail-corrected spectrum, the dashed black line for the uncorrected case. (Note the logarithmic y-axis, varying over 3 orders of magnitude).

Studies based on reanalysis meteorology have also provided a diverse picture of mean age trends (e.g., Diallo et al., 2012; Monge-Sanz et al., 2012; Ploeger et al., 2015a) which, however, is partly related to the different trend periods considered.

In view of such diversity in past published mean age trends, the agreement between the "long-term" trends for 1989–2015 in the lower stratosphere (below about 30 hPa) presented here for the newest three reanalyses is remarkable (Fig. 7). In general,

5 the mean age trend is negative throughout the lower stratosphere with largest values in the tropical and SH lower stratosphere, indicating an accelerating shallow BDC branch. These negative long-term mean age trends in current reanalyses appear largely consistent with mean age trends from climate models over even longer periods (century), representing the forced response of the BDC to increasing greenhouse gas levels. Consistent with these negative mean age trends and the related shift of the age spectrum peak (modal age) towards shorter transit times (e.g., Fig. 9), most current reanalyses show an increasing tropical mass

10 flux (Abalos et al., 2015) and decreasing residual circulation transit times (Birner et al., in preparation). However, Abalos et al.





(2015) further pointed out that residual circulation trends may depend critically on the method used for calculating upwelling velocities.

Above about 30 hPa mean age trends and related circulation changes depend more strongly on the respective reanalysis considered. At these upper levels ERA–Interim shows ageing in the NH, which is not included in the other two reanalyses.

Hence, long-term changes in the deep BDC branch appear less robustly represented in the reanalyses. The NH ageing in ERA–Interim turns out to be related to a strengthening age spectrum tail indicating changes in recirculation. The evidence for recirculation changes agrees with the results of Ploeger et al. (2015a), showing that atmospheric mixing processes substantially affect mean age trend patterns, with strongest effects on shorter time scales like decades.

On shorter periods of around a decade the reanalysis age of air trends show substantial differences. For the pre–2000 period,

trends in the NH above about 600 K (approximately 24 km) are not consistent (Fig. 7), due to insignificant ERA–Interim trends (Fig. 7d). The strong negative mean age trends in the SH and tropics, on the other hand, consistently emerge from all reanalyses and likely represent the combined effects of ozone depleting substances to decrease mean age (Polvani et al., 2018) and of volcanic aerosol from the Pinatubo eruption to increase mean age at the beginning of the 1989–2001 period (Diallo et al., 2017). In particular regarding the effect of Pinatubo aerosol, differences in the response of the BDC and related differences in

age of air between the reanalyses are evident from the middle latitude mean age time series in Fig. 11b. The increase in age after the Pinatubo eruption is very strong in JRA–55, weaker in ERA–Interim, and absent in MERRA–2. These differences point to different representations of stratospheric volcanic aerosol in the reanalyses causing the differences in decadal BDC trends. In ERA–Interim and JRA–55 the effects of stratospheric volcanic aerosol are only included by assimilation of observed temperature and wind data, as discussed in more detail by Diallo et al. (2017), whereas MERRA–2 in addition assimilates

aerosol optical depth (Fujiwara et al., 2017). Also the effect of ODS is very likely differently represented in the reanalyses due to the usage of different ozone products. ERA–Interim uses an ozone climatology, JRA–55 time-varying ozone fields from another model simulation, and MERRA–2 interactive ozone (see Sect. 3). Hence, differences in the related effects on the BDC are to be expected.

For the post–2000 period, age of air trends are even less consistent and no clear common circulation change emerges from

comparison of the different reanalyses (Fig. 7). Remarkably, largest differences between the reanalysis trends occur for the most recent period. This is against the general expectation that a larger consistency between reanalyses emerges for more recent periods, when observational data and the assimilation procedures are more consistent. The inconsistency in post–2000 trends is mainly related to the fact that JRA–55 and MERRA–2 show opposite mean age trends during 2002–2015 throughout the stratosphere (Fig. 7h, i), and that only ERA–Interim shows a hemispheric dipole pattern (Fig. 7g). The hemispheric dipole

pattern in mean age changes in ERA–Interim is related to changes in the age spectrum tail (increasing tail in the NH, decreasing tail in the SH), pointing to changes in recirculation in agreement with Ploeger et al. (2015a). As the hemispheric dipole age trend pattern has been related recently to a southward circulation shift by about 5 degrees (Stiller et al., 2017), post–2000 age trend differences between the reanalyses could be caused by differences in the representation of this circulation shift.

Comparison to the recently published reanalysis mean age trends by Chabrillat et al. (2018) may reveal interesting differ-

ences regarding the representation of the BDC between kinematic transport models as used in their study (BASCOE model) and



diabatic transport models as used here (CLaMS). Regarding the climatology, mean age from MERRA–2 shows substantially higher values than the other reanalyses for both diabatic and kinematic transport (see Fig. 1, and Chabrillat et al., 2018, Fig. 3). The differences in diabatic heating rates, with weaker tropical upwelling for MERRA–2 (Fig. 5), are qualitatively consistent with these differences in mean age in the diabatic model calculation. The similar finding by Chabrillat et al. (2018) suggests

that similar differences are included in the reanalysis vertical winds which drive their kinematic model calculations.

However, as noted earlier, the differences in heating rates alone cannot fully explain the simulated mean age differences as shown in the following. Figure 13 shows ERA–Interim climatological mean age and residual circulation transit times (RCTT), and differences to the other reanalyses. Here, RCTTs are calculated from the diabatic residual circulation in isentropic coordinates, as described in Sect. 2. Mean age from MERRA–2 is older compared to ERA–Interim by more than 2 years throughout

most parts of the stratosphere (Fig. 13c), whereas differences in RCTTs are below 0.5 years (Fig. 13f). Hence, differences in the representation of mixing processes must play a role.

The net mixing effect on mean age (ageing by mixing), calculated as the difference between mean age and RCTT following Garny et al. (2014) and further described in Sect. 2, is shown in Fig. 13g–i. This ageing by mixing is about 2 years larger for MERRA–2 compared to ERA–Interim in most parts of the lower stratosphere, and clearly explains the large difference

in mean age between the reanalyses. However, ageing by mixing depends on both local mixing characteristics in the lower stratosphere and the transit time of tropical upwelling, which controls the time scale for mixing to affect the ascending air (see Garny et al., 2014; Ploeger et al., 2015a). Hence, the differences in the ageing by mixing between ERA–Interim and MERRA–2 may be caused by either differences in local mixing (eddy diffusivity) or by differences in RCTTs. From a tropical leaky pipe model perspective ageing by mixing is even linearly related to RCTT, with longer RCTT causing larger ageing by

mixing (Neu and Plumb, 1999; Dietmüller et al., 2018). In percent the differences in RCTTs between MERRA–2 and ERA–Interim in the tropical lower stratosphere is large (about 50%, see blue contours in Fig. 13f), and likely causes a related difference in ageing by mixing. Overall, the larger ageing by mixing for MERRA–2 compared to ERA–Interim clearly causes the differences in mean age. Deeper insight into potential differences in local mixing characteristics could be gained by a reanalysis inter-comparison of effective diffusivity (e.g., Haynes and Shuckburgh, 2000; Abalos et al., 2017).

For the long-term mean age trends (1989–2015) both diabatic and kinematic transport representations result in very similar results for all three reanalyses, as evident from comparison of Fig. 7 with Fig. 12 of Chabrillat et al. (2018). This good agreement regarding long-term trends again points to a robust representation of the effect of greenhouse gas induced warming in forcing trends in the BDC. Regarding decadal changes over shorter periods, however, results from diabatic and kinematic transport models show more substantial differences. Strongest differences emerge for the pre–2000 period for ERA–Interim

and for the post-2000 period for MERRA–2. For the former case, diabatic transport in CLaMS results in negative mean age trends throughout the SH and tropical stratosphere (Fig. 7d), whereas kinematic transport in BASCOE results in positive trends in these regions (Chabrillat et al., 2018, Fig. 12). As pre–2000 decadal trends are likely controlled by the effects of ozone depleting substances and Pinatubo volcanic aerosol on the BDC, the differences in age trends for this period point to differences in the representation of these processes between diabatic and kinematic transport. In particular differences in the effect of volcanic

aerosol (e.g., due to the Pinatubo eruption in June 1991), with increasing mean age for diabatic transport and no significant



age changes for kinematic transport seem to be critical, and have also been noted by Chabrillat et al. (2018). For the latter case (post-2000 period for MERRA–2), diabatic transport in CLaMS results in negative trends throughout the stratosphere (Fig. 7i), whereas kinematic transport in BASCOE results in positive age trends (Chabrillat et al., 2018, Fig. 12). Furthermore, less striking differences between diabatic and kinematic mean age trends emerge for MERRA–2 during the pre–2000 period, where kinematic age trends are more negative compared to diabatic trends (Fig. 7f and Chabrillat et al., 2018, Fig. 12). Weak differences also occur for ERA–Interim during 2002–2015, where the NH ageing appears stronger in the diabatic age trends (Fig. 7g and Chabrillat et al., 2018, Fig. 12). The causes for the differences regarding decadal trends in the BDC on shorter periods between diabatic and kinematic transport representations, as well as between the different reanalyses, are unclear so far, but appear to be a promising subject for future analysis aiming to improve consistency between the reanalyses.

## 7 Conclusions

We compared stratospheric mean age and the full age spectrum from simulations with the diabatic CLaMS model driven by different reanalyses (ERA–Interim, JRA–55, MERRA–2) to investigate the robustness of representing the climatology, seasonality, and trends of the Brewer-Dobson circulation in current generation reanalysis data sets. Considering the full (time-dependent) age spectrum in a data inter-comparison is novel and provides clearer insights into circulation differences than considering mean age. Climatological mean age differs significantly between the different reanalyses, with JRA–55 showing the youngest age and MERRA–2 showing the oldest age throughout almost the entire stratosphere. The substantially older mean age for MERRA–2 appears to be related to a more pronounced age spectrum tail, indicating a stronger effect of recirculation of air into the tropics. A comparison of residual circulation and mixing effects on mean age confirms that the net mixing effect (including recirculation) is necessary for explaining the mean age differences between MERRA–2 and ERA–Interim. The seasonality in the BDC is robustly represented, with very similar mean age seasonality and similar seasonal age spectrum peaks emerging for all reanalyses. Comparison to balloon-borne mean age observations reveals a similarly large spread in simulated and observed mean age and allows no clear conclusions to be drawn regarding the reliability of the different reanalyses.

In particular, long-term trends in the lower stratosphere (below about 30 hPa) during the 1989–2015 period are robustly represented in the reanalyses showing mainly decreasing mean age, strongest in the SH and tropical lower stratosphere. Related to this mean age decrease is a robust shift of the age spectrum towards shorter transit times and an increase in the fraction of young air masses. These long-term age of air changes from reanalyses resemble results from climate model simulations over even longer periods, which simulate an acceleration of the shallow branch of the stratospheric BDC as the forced response to increasing greenhouse gas levels. At upper levels (above 30 hPa), mean age changes in the reanalyses appear less robust, pointing to a less robust representation of changes in the deep BDC branch.

For shorter periods of about a decade (here 1989–2001 and 2002–2015), age of air changes are more diverse and depend on the specific reanalysis considered. These decadal age changes may even disagree in sign globally for certain periods. Moreover, the hemispheric asymmetric dipole in mean age trends for 2002–2015, as viewed by satellite observations, only emerges for

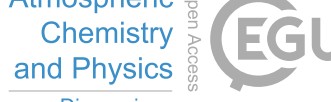



**Figure 13.** Mean age from ERA–Interim (a), and differences to JRA–55 (b) and MERRA–2 (c), for the 1989–2013 climatological mean. The same for residual circulation transit time RCTT is shown in (d–f), and for ageing by mixing in (g–i). Thin black solid contours show the climatology values for the respective reanalysis (e.g., ERA–Interim mean age in (a), MERRA–2 mean age in (c)). Thin black dashed lines show pressure levels, the thick black solid line the (WMO) tropopause. The blue contours in (e–f) show relative differences for RCTTs in percent (solid for positive, dashed for negative differences).

ERA–Interim. Hence, decadal variability in the Brewer-Dobson circulation and the various factors involved (e.g., QBO, ENSO, ODS, volcanic aerosol) turn out to be not robustly represented in current generation reanalyses.

*Author contributions.* FP carried out the ERA–Interim and JRA–55 driven model simulations and the data analysis. BL and XY prepared the reanalysis data. EC carried out the MERRA–22 driven simulation. PK, MT, XY and BL contributed contributed code for the analysis.



BL, TB, PK and MD contributed to the design of the analysis. AE provided the observational mean age data. MD, PK, TB, MT, MR, EC, BL, XY provided helpful discussion and comments. FP wrote the manuscript with contributions from all co-authors.

*Acknowledgements.* We thank Marta Abalos, Aurelien Podglajen and Gebhard Günther for helpful discussion. We further thank Nicole Thomas for programming support, and the ECMWF, the NASA, and the Japanese Meteorological Agency for providing the reanalysis data.

5 The CLaMS model data may be requested from the corresponding author (f.ploeger@fz-juelich.de). This study was funded by the Helmholtz Association under grant VH-NG-1128 (Helmholtz Young Investigators Group A–SPECi).



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
