# Peer review of "How robust are stratospheric age of air trends from different reanalyses?"

_Atmospheric Chemistry and Physics, 2018_

## Referee Comment (RC1) · Anonymous Referee #3 · 11 Feb 2019

The paper presents an intercomparison of mean stratospheric age of air and age spectrum for three modern reanalyses using the diabatic model CLaMS. The climatology, seasonality and long-term trends are evaluated, and the results are compare to observations and a previous reanalysis study based on a kinematic model. A large spread in the climatological values is pointed out, which is however comparable to the range of uncertainty in observations. The seasonality is similar in all reanalyses. The long-term trends are qualitatively consistent over 1989-2015 but less so over shorter (decadal) periods. Overall, the results confirm a long-term acceleration of the BDC consistent with model predictions in response to increasing greenhouse gases.

The topic is of high interest, the paper is timely, comprehensive and very well written, and the results are clearly presented. I recommend publication. The only comments I

have, listed below, are mostly technical.

- P7 L1: Perhaps you could briefly comment the impacts of choosing a given spin-up year.

- P9 L28: remove 'complete'

- P11 L1-2 and P12 L13-14: Could you be more specific on what is meant by 'the tropics-extratropics transition is more dilute'? Do you mean the smoother mean age latitudinal gradients or less contrast tropics/extratropics in the age spectrum amplitude values?

- P13 L2: Remove 'further'

- P14 L1: Fig. 7 a-c (add a-c)

- Fig. 8 caption: add 'annual mean', otherwise one is tempted to compare with Fig. 3.

- P17 L5: 'chemical and radiative': Abalos et al. (2019) JGR (https://doi.org/10.1029/2018JD029301) show that the negative mean age trends in the SH are attributed to the ozone hole.

- P19 7-9: This sentence is confusing, it would be better to compare different reanalyses over the same period, not two reanalyses over two different periods.

- P20 L17: no conclusion is possible with regards to which reanalysis . . .

- P22 L3: the qualitative agreement

- P24 L4-5: Unclear sentence: are differences in vertical winds consistent with differences in heating rates (among reanalyses)?

- P24 L24: The year of the second reference should be 2016

- P25 L12: robustness in the representation of . . .

- P25 L15: than considering mean age alone.

---

## Referee Comment (RC2) · Anonymous Referee #2 · 11 Feb 2019

This study presents an inter-comparison of three reanalyses with respect to the simulation of the Brewer Dobson circulation (BDC). Age of air (AoA) was used as a metric for the BDC and estimated by using the Chemical Lagrangian Model of the Stratosphere (CLaMS) driven by the respective reanalyses. Beside stratospheric mean age, the full age spectrum was calculated to investigate the robustness of the representation of climatology, seasonality, and trends of the BDC in reanalysis data sets.

The paper is well written and clearly structured and provides a comprehensive analysis of the stratospheric BDC in different reanalysis data sets. It is suitable for publication in ACP after minor revision.

General comments:

[Figure]

1. The first general comment refers to the usage of terms like "older age". It is a similar expression like "warmer temperatures (instead of higher temperatures)" and used in common speech (everyone knows clearly, what is meant), although it is a false use. I would suggest to replace this notion by "higher age" or "older air", that are in form and content correct descriptions. This replacement should be done throughout the text.

2. Some of the coloured figures are hard to read, because the description of contour lines are too small or the colour is too dark (Fig. 6c, dark blue contour lines on dark grey colour), Fig 7d-f, Fig. 13.

3. Please explain how you estimated the AoA trends and significance. Did you consider correlations between near by values, when you calculated the trend patterns?

Minor comments:

Page 1, line 2: "analyze" or "analyse" ?, please be consistent throughout the text.

Page 2, line 27: Please, do not cite publications, which are in preparation and not available to the reader.

Page 9, line 25: Did you mean Fig. 2c? Fig. 2f does not exist.

Page 22, line 10: "Birner, personal communication"

Page 24, line 4: The corresponding finding by Chabrillat ...

Page 28, line 32-33: Please cite correctly: Chabrillat, S., Vigouroux, C., Christophe, Y., Engel, A., Errera, Q., Minganti, D., Monge-Sanz, B. M., Segers, A., and Mahieu, E.: Comparison of mean age of air in five reanalyses using the BASCOE transport model, Atmos. Chem. Phys., 18, 14715-14735, https://doi.org/10.5194/acp-18-14715-2018, 2018.

Page 30, lines 24-26: Please cancel the citation "Hauck, M ..."

Page 32, line 2: Please cancel "in preparation".

---

## Referee Comment (RC3) · Eric Ray (Referee) · 11 Feb 2019

This paper uses the CLaMS model driven by meteorology from three different reanalysis products to investigate stratospheric age of air trends over the past few decades. The use of pulsed tracer releases allows the computation of age spectra at all locations and times in the stratosphere and this provides a powerful additional diagnostic to give insight into age of air trends beyond mean age. The paper is well written and the results are a significant contribution to our current understanding, or lack of understanding, of the recent variability in the circulation and mixing in the stratosphere. The lack of agreement in the decadal variability of the stratospheric age of air produced by the different reanalysis products is disturbing, especially in the most recent period when, as the authors mention, we would expect the reanalyses to come into better agreement. This

study adds to a growing body of literature pointing out important discrepancies in the representation of transport and mixing in the stratosphere and troposphere between reanalyses.

I recommend publication with consideration of the grammatical comments below.

Specific comments:

Pg. 1, line 17: change 'with' to 'of'

Pg. 2, line 17: add comma after 'scales'

Pg. 3, line 14: add comma after 'periods'

Pg. 3, line 25: add comma after 'time'

Pg. 4, lines 6-7: 'The results presented here are based…'

Pg. 4, line 14: change 'against' to 'compared to'

Pg. 5, lines 21-22: 'As the full age spectrum is the probability distribution of transit times, it is in general normalized to unity.'

Pg. 5, line 27: remove 'for' before MERRA-2. Should change 'stronger' to something else, perhaps 'flatter'.

Pg. 6, line 30: '…analysis to begin in 1979 (in 1980 for MERRA-2)…'

Pg. 7, line 8: remove 'finally'

Pg. 9, line 25: there is no Fig. 2f

Pg. 12, line 2: remove 'already', change 'clearer' to 'more clear'

Pg. 12, line 4: '…consistent with the comparison of reanalysis…'

Pg. 12, line 14: remove 'also'

Pg. 14, line 2: add 'a-c' after 'Fig. 7'

Pg. 14, line 6: add 'to be' after 'appear'

Pg. 15, line 3: add 'a-c' after 'Fig. 7'

Pg. 15, line 8: '…age spectrum can shed more light on the processes involved during each period.'

Pg. 15, line 9: change 'lowest' to 'lower'

Pg. 15, line 28: change to 'particularly'

Pg. 16, line 4: '…taken as representative of long-term trends.'

Pg. 16, line 8: change 'long' to 'longer'

Pg. 17, line 2: add 'the' before '2-3 years'

Pg. 19, line 4: change 'stronger' to 'more'

Pg. 19, line 16: add comma after 'SH'

Pg. 20, line 17: add 'regarding' after 'possible'

Pg. 20, line 29: 'This longer tail decay time scale causes the finite tail…'

Pg. 20, line 30: change 'has' to 'to have'

Pg. 24, line 20: add comma after 'percent'

Pg. 25, line 19: change 'explaining' to 'to explain'

Pg. 26, line 2: '…to not be robustly…'

---

## Author Comment (AC1) · 1 Apr 2019

We thank the Reviewer for her/his positive judgement of the manuscript and the good comments. In the following, we address all comments and questions raised (Reviewer's comments in italics). Text changes in the manuscript are highlighted in color (except minor wording changes).

**General comments:**

*The paper presents an intercomparison of mean stratospheric age of air and age spectrum for three modern reanalyses using the diabatic model CLaMS. The climatology, seasonality and long-term trends are evaluated, and the results are compare to observations and a previous reanalysis study based on a kinematic model. A large spread in*

[Figure]

*the climatological values is pointed out, which is however comparable to the range of uncertainty in observations. The seasonality is similar in all reanalyses. The long-term trends are qualitatively consistent over 1989-2015 but less so over shorter (decadal) periods. Overall, the results confirm a long-term acceleration of the BDC consistent with model predictions in response to increasing greenhouse gases. The topic is of high interest, the paper is timely, comprehensive and very well written, and the results are clearly presented. I recommend publication. The only comments I have, listed below, are mostly technical.*

**Minor and Technical comments:**

P7, L1: *Perhaps you could briefly comment the impacts of choosing a given spin-up year.*

In general, the truncation of the age spectrum at a transit time of 10 years causes a young bias in mean age, as discussed in the paper (e.g., P5, L15ff). The age spectra do not include spin-up effects for most parts of the analysis, as we consider the period from 1989 on, which is after 10 preceding years of simulation. Only for MERRA-2, which starts only in 1980, the age spectrum tail (between 9-10 years transit time) includes a remaining spin-up effect in the year 1989, from the preceding spin-up phase using perpetuum 1979 conditions, as explained on P7, L2.

To enable comparison of model mean age to balloon observations in the years before 1989 in Fig. 11, a 10 year spin-up is preceding the main simulation with repeating conditions of the year 1979. All data points before 1989 in Fig. 11 are therefore influenced by this spin-up, with a weaker effect when approaching 1989. This influence only occurs for the comparison of Fig. 11 and is now stated also on P20, L15.

P9, L28: *Remove "Complete".*

Done.

P11, L1-2, P12, L13-14: *Could you be more specific on what is meant by "the tropics-*

*extratropics transition is more dilute"? Do you mean the smoother mean age latitudinal
gradients or less contrast tropics/extratropics in the age spectrum amplitude values?*

Thanks for pointing this unclear formulation out. It is particularly the transition between
the tropical and extratropical age spectra which is more dilute, and this is now clearly
stated in the revised manuscript.

P13, L2: *Remove "further".*

Done.

P14 L1: *Fig. 7 a-c (add a-c).*

Done.

Fig. 8 caption: *add "annual mean", otherwise one is tempted to compare with Fig. 3.*
Done.

P17 L5: '*"chemical and radiative": Abalos et al. (2019) JGR
(https://doi.org/10.1029/2018JD029301) show that the negative mean age trends
in the SH are attributed to the ozone hole.*

Thanks for emphasizing this point. It is clarified in the revised manuscript, and the new
Abalos et al. paper is cited.

P19 7-9: *This sentence is confusing, it would be better to compare different reanalyses
over the same period, not two reanalyses over two different periods.*

The point here is that the differences between different reanalyses are larger for the
shorter periods considered. We rephrased the respective sentence to clarify that.

P20 L17: *no conclusion is possible with regards to which reanalysis*

We would keep the former formulation as we think it is clearer in making the point that
we can not say which reanalysis is most realistic.

P22 L3: *the qualitative agreement.*

Done.

P24 L4-5: *Unclear sentence: are differences in vertical winds consistent with differences in heating rates (among reanalyses)?*

The sentence should just hypothesize that differences in vertical winds could be similar to differences in heating rates, because the differences in mean age from the simulations driven with either vertical velocity are similar. We rephrased the sentence to clarify that.

P24 L24: *The year of the second reference should be 2016*

Changed.

P25 L12: *robustness in the representation of*

Changed.

P25 L15: *than considering mean age alone.*

Changed.

---

## Author Comment (AC2) · 1 Apr 2019

We thank the Reviewer for her/his positive judgement of the manuscript and the good comments. In the following, we address all comments and questions raised (Reviewer's comments in italics). Text changes in the manuscript are highlighted in color (except minor wording changes).

**General comments:**

*This study presents an inter-comparison of three reanalyses with respect to the simulation of the Brewer Dobson circulation (BDC). Age of air (AoA) was used as a metric for the BDC and estimated by using the Chemical Lagrangian Model of the Stratosphere (CLaMS) driven by the respective reanalyses. Beside stratospheric mean age, the*

[Figure]

*full age spectrum was calculated to investigate the robustness of the representation of climatology, seasonality, and trends of the BDC in reanalysis data sets. The paper is well written and clearly structured and provides a comprehensive analysis of the strato-spheric BDC in different reanalysis data sets. It is suitable for publication in ACP after minor revision.*

*1. The first general comment refers to the usage of terms like "older age". It is a similar expression like "warmer temperatures (instead of higher temperatures)" and used in common speech (everyone knows clearly, what is meant), although it is a false use. I would suggest to replace this notion by "higher age" or "older air", that are in form and content correct descriptions. This replacement should be done throughout the text.*

Thanks for pointing this ill formulation out. We changed it according to the Reviewer's suggestion throughout the manuscript.

*2. Some of the coloured figures are hard to read, because the description of contour lines are too small or the colour is too dark (Fig. 6c, dark blue contour lines on dark grey colour), Fig 7d-f, Fig. 13.*

Thanks for this remark. We agree that the choice of line colours is not optimal for some of the panels (e.g., Fig. 6c, Fig. 7e). Unfortunately, for such multi-panel figures it is almost impossible to find colour schemes which are optimal for all panel. The colour choices in the manuscript are already the result of trying to optimize readability for all sub-panels with using one common colour scheme.

*3. Please explain how you estimated the AoA trends and significance. Did you consider correlations between near by values, when you calculated the trend patterns?*

We calculated the trends simply from linear regression of monthly mean time series after deseasonalizing (by subtracting the mean annual cycle). The significance (e.g., in Fig. 7) is measured in multiples of standard deviation of the linear trend (as stated in the figure caption). We compared our results also to results from the more sophisticated

regression method by Diallo et al. (2017), but found no significant differences.

**Minor comments:**

Page 1, line 2: *"analyze" or "analyse" ?, please be consistent throughout the text.*

We changed it everywhere to "analyse".

Page 2, line 27: *Please, do not cite publications, which are in preparation and not available to the reader.*

The publication by Hauck et al. is now accepted abd available, and the reference is updated to ACP.

Page 9, line 25: *Did you mean Fig. 2c? Fig. 2f does not exist.*

Thanks for the careful reading. The respective remark actually refers to all three sub-panels of Fig. 2, and we simply removed the "f".

Page 22, line 10: *"Birner, personal communication"*

Changed.

Page 24, line 4: *The corresponding finding by Chabrillat ...*

The entire sentence has been rephrased, also including the comments by Reviewer 3.

Page 28, line 32-33: *Please cite correctly: Chabrillat, S., Vigouroux, C., Christophe, Y., Engel, A., Errera, Q., Minganti, D., Monge-Sanz, B. M., Segers, A., and Mahieu, E.: Comparison of mean age of air in five reanalyses using the BASCOE transport model, Atmos. Chem. Phys., 18, 14715-14735, https://doi.org/10.5194/acp-18-14715-2018, 2018.*

Corrected.

Page 30, lines 24-26: *Please cancel the citation "Hauck, M ..."*

The citation by Hauck et al. has been updated now to "accepted for publication in ACP".

Page 32, line 2: *Please cancel "in preparation".*

The publication by Podglajen and Ploeger also has been updated for the correct ACP
reference.

---

## Author Comment (AC3) · 1 Apr 2019

We thank Eric Ray for his positive judgement of the manuscript, the careful reading and detailed specific comments. In the following, we address all his comments (Reviewer's comments in italics). Text changes in the manuscript are highlighted in color (except minor wording changes).

**General comments:**

*This paper uses the CLaMS model driven by meteorology from three different reanalysis products to investigate stratospheric age of air trends over the past few decades. The use of pulsed tracer releases allows the computation of age spectra at all locations and times in the stratosphere and this provides a powerful additional diagnostic to give*

[Figure]

*insight into age of air trends beyond mean age. The paper is well written and the results are a significant contribution to our current understanding, or lack of understanding, of the recent variability in the circulation and mixing in the stratosphere. The lack of agreement in the decadal variability of the stratospheric age of air produced by the different reanalysis products is disturbing, especially in the most recent period when, as the authors mention, we would expect the reanalyses to come into better agreement. This study adds to a growing body of literature pointing out important discrepancies in the representation of transport and mixing in the stratosphere and troposphere between reanalyses.*

*I recommend publication with consideration of the grammatical comments below.*

**Specific comments:**

Pg. 1, line 17: *change "with" to "of"*

Changed.

Pg. 2, line 17: *add comma after "scales"*

Done.

Pg. 3, line 14: *add comma after "periods"*

Done.

Pg. 3, line 25: *add comma after "time"*

Done.

Pg. 4, lines 6-7: *"The results presented here are based"*

Changed.

Pg. 4, line 14: *change "against" to "compared to"*

Changed.

Pg. 5, lines 21-22: *"As the full age spectrum is the probability distribution of transit times, it is in general normalized to unity."*

Changed.

Pg. 5, line 27: *remove "for" before MERRA-2. Should change 'stronger' to something else, perhaps "flatter".*

Changed. ("stronger" changed to "more pronounced").

Pg. 6, line 30: *"analysis to begin in 1979 (in 1980 for MERRA-2)"*

Changed.

Pg. 7, line 8: *remove "finally"*

Done.

Pg. 9, line 25: *there is no Fig. 2f*

Thanks for recognizing this. The respective statement actually refers to all 3 sub-panels of Fig. 2 and we simply removed the "f".

Pg. 12, line 2: *remove "already", change "clearer" to "more clear"*

Done.

Pg. 12, line 4: *"consistent with the comparison of reanalysis"*

Changed.

Pg. 12, line 14: *remove "also"*

Done.

Pg. 14, line 2: *add "a-c" after "Fig. 7"*

Done.

Pg. 14, line 6: *add "to be" after "appear"*

Done.

Pg. 15, line 3: *add "a-c" after "Fig. 7"*

Done.

Pg. 15, line 8: *"age spectrum can shed more light on the processes involved during each period."*

Changed.

Pg. 15, line 9: *change "lowest" to "lower"*

Changed.

Pg. 15, line 28: *change to "particularly"*

Changed.

Pg. 16, line 4: *"taken as representative of long-term trends."*

Changed.

Pg. 16, line 8: *change "long" to "longer"*

Done.

Pg. 17, line 2: *add "the" before "2-3 years"*

Done.

Pg. 19, line 4: *change "stronger" to "more"*

Done.

Pg. 19, line 16: *add comma after "SH"*

Done.

Pg. 20, line 17: *add "regarding" after "possible"*

Done.

Pg. 20, line 29: *"This longer tail decay time scale causes the finite tail"*

Done.

Pg. 20, line 30: *change "has" to "to have"*

Done.

Pg. 24, line 20: *add comma after "percent"*

Done.

Pg. 25, line 19: *change "explaining" to "to explain"*

Changed.

Pg. 26, line 2: *"to not be robustly"*

Changed.
* * *